# HIV-1 Subtype C Drug Resistance Mutations in Heavily Treated Patients Failing Integrase Strand Transfer Inhibitor-Based Regimens in Botswana [note 1]

**DOI:** 10.3390/v13040594

**Published:** 2021-03-31

**Authors:** Kaelo K. Seatla, Dorcas Maruapula, Wonderful T. Choga, Tshenolo Ntsipe, Nametso Mathiba, Mompati Mogwele, Max Kapanda, Bornapate Nkomo, Dinah Ramaabya, Joseph Makhema, Mompati Mmalane, Madisa Mine, Ishmael Kasvosve, Shahin Lockman, Sikhulile Moyo, Simani Gaseitsiwe

**Affiliations:** 1Botswana Harvard AIDS Institute Partnership, Gaborone 0000, Botswana; dmaruapula@gmail.com (D.M.); wchoga@bhp.org.bw (W.T.C.); nmathiba@bhp.org.bw (N.M.); mogwelem@yahoo.com (M.M.); jmakhema@bhp.org.bw (J.M.); mmmalane@bhp.org.bw (M.M.); shahin.lockman@gmail.com (S.L.); smoyo@bhp.org.bw (S.M.); sgaseitsiwe@gmail.com (S.G.); 2School of Allied Health Professions, Faculty of Health Sciences, University of Botswana, Gaborone 0000, Botswana; kasvosvei@ub.ac.bw; 3Division of Human Genetics, Department of Pathology, Faculty of Health Sciences, University of Cape Town, Cape Town 7925, South Africa; 4National Health Laboratory, Ministry of Health & Wellness, Gaborone 0000, Botswana; tntsipe@bhp.org.bw (T.N.); madisamine0@gmail.com (M.M.); 5Botswana Ministry of Health and Wellness, Gaborone 0000, Botswana; maxkapanda@gmail.com (M.K.); bnkomo@gov.bw (B.N.); dramaabya@gov.bw (D.R.); 6Division of Infectious Diseases, Brigham and Women’s Hospital, Harvard Medical School, Boston, MA 02115, USA; 7Department of Immunology & Infectious Diseases, Harvard T.H. Chan School of Public Health, Boston, MA 02115, USA

**Keywords:** Botswana, dolutegravir, resistance, mutations, HIV-1C, integrase inhibitors

## Abstract

There are limited real-world mutational and virological outcomes data of treatment-experienced persons diagnosed with HIV-1 subtype C (HIV-1 C) who are failing Integrase Strand Transfer Inhibitor-based regimens. Requisition forms sent for HIV-1 genotypic resistance testing (GRT) between May 2015 and September 2019 were reviewed and participants experiencing virologic failure while on dolutegravir (DTG) or raltegravir (RAL) cART at sampling recruited. Sanger sequencing of the HIV-1 Pol gene was performed from residual plasma samples and drug resistance mutational (DRM) analysis performed using the Stanford University HIV drug resistance database. 40 HIV-1C integrase sequences were generated from 34 individuals, 24 of whom were on DTG cART, three on RAL cART and seven on an unknown (DTG or RAL)-anchored cART at time of GRT. 11/34 (32%) individuals had DRMs to DTG and other integrase inhibitors. 7/11 (64%) patients had exposure to a RAL-based cART at the time of sampling. Out of the 11 individuals with DRMs, one (9%) had 2-class, 6 (55%) had 3-class, and 4 (36%) had 4-class multidrug-resistant HIV-1C. 7/11 individuals (64%) are currently virologically suppressed. Of the four individuals not virologically suppressed, three had extensive DRMs involving 4-classes of ARV drugs and one individual has demised. Resistance to DTG occurs more often in patients exposed to RAL cART. Individuals with 4-class DRMs plus integrase T97 and E157Q mutations appear to have worse outcomes. There is a need for frequent VL monitoring and GRT amongst treatment-experienced HIV-1C diagnosed individuals.

## 1. Introduction

Dolutegravir (DTG) is a second-generation integrase strand transfer inhibitor (INSTI) with superior efficacy, a higher genetic barrier to resistance, and a better safety profile than raltegravir (RAL), efavirenz, darunavir-ritonavir (DRV\r) combination antiretroviral therapy’s (cART) [1,2]. It is recommended as part of the first-line cART by multiple HIV treatment guidelines [3,4,5,6]. However, the emergence of antiretroviral (ARV) resistant HIV-1 is a rising global health threat [7] due to risk of onward transmission of drug-resistant HIV-1 variants [8,9,10] in addition to failure to achieve virological suppression with subsequent increase in morbidity and mortality [11,12]. Globally about 38 million people are living with HIV (PLWH) [13]. Botswana, with a population of about 2.2 million, has an adult HIV prevalence of 20.7% and about 310,000 PLWH on ART [14,15]. HIV-1 subtype C (HIV-1C) predominates in Botswana and the region. Botswana was amongst the first low/middle income country (LMIC) to adopt INSTIs such as RAL and DTG to its mature free ART programme for highly treatment-experienced individuals, and also adopted DTG in first line-therapy in 2016 [4]. However, there is limited ‘real-life’ clinical and virologic outcomes programmatic data on PLWH diagnosed with HIV-1C failing DTG/RAL based regimens.

Highly treatment-experienced PLWH previously failing a RAL-based cART regimen for prolonged periods are prone to develop virological failure (VF) when switched to DTG based cART due to cross-resistance and accumulation of more drug-resistant mutations (DRMs) that reduce DTG’s efficacy [16]. INSTI-naive highly treatment-experienced patients failing DTG cART have been found to have a virus with DRMs including G118R, D67N; H51H/Y, G118R, E138E/K, and less commonly R263R/K, V260I, R263R, N155H, G118R, and E138E [17]. These cases were mainly from clinical trials dominated by non-HIV-1C viruses [18,19,20,21]. However, there is limited information on the selection of INSTI DRMs and subsequent treatment outcomes on INSTI-based cART, especially amongst HIV-1C infected individuals.

We performed a comprehensive clinical and drug resistance genotypic characterization of HIV-1C from PLWH experiencing VF on DTG and/or RAL -based cART in Botswana.

## 2. Materials and Methods

### 2.1. Study Setting

HIV care, including cART, is provided free of charge to all citizens diagnosed with HIV in Botswana. RAL and DTG have been available in the Botswana national HIV treatment program cART since late 2008 and early 2016 respectively [2].

### 2.2. Selection of Study Population

We reviewed paper-based laboratory requisition forms sent for genotypic resistance testing on HIV-1 diagnosed adults (>18 years) who experienced virologic failure (generally two or more viral loads (VL) greater than 400 copies/mL while on cART, as per standard of care. These individuals were accessing public health facilities across Botswana as part of their routine HIV clinical care between May 2015 and September 2019. 

We included participants whose laboratory requisition forms indicated that they were on DTG, RAL, or ‘INSTI’ cART at the time of the request. Laboratory requisition forms were accompanied by plasma biospecimens which were sent to a central laboratory, the Botswana Harvard HIV Reference Laboratory (BHHRL) which is SADCAS ISO 15189 accredited.

### 2.3. Clinical and Laboratory Methods Description

#### 2.3.1. Clinical Data Extraction 

We reviewed electronic and paper-based medical records to extract demographics, prior/current/subsequent ART regimens, HIV-1 RNA results, and documentation of poor adherence. 

#### 2.3.2. Viral Load Quantification

HIV-1 VL was quantified by either Abbott m2000sp/Abbott m2000rt platform (Wiesbaden, Germany), Cobas TaqMan/Cobas Ampliprep HIV-test (Roche Molecular Systems, Branchburg, NJ, USA) or Aptima HIV-1 Quant assay on Panther Systems (Hologic inc., San Diego, CA, USA) at BHHRL or district-based laboratories. VF was defined as two consecutive VL greater than 400 copies/mL and virologic suppression as a viral load < 400 copies/mL as per national ART guidelines. Because of the various VL detection platforms used and changes in the national VL reporting guidelines, some of the VL data was reported as <400, <50, <40 and <25 copies/mL.

#### 2.3.3. Genotypic Resistance Testing (GRT)

Using residual plasma samples from individuals undergoing routine genotypic resistance testing, we amplified and Sanger sequenced the integrase (IN) region and/or reverse transcriptase (RT) and protease (PR) genes. We included testing of multiple plasma specimens that were identified to come from one individual over time and which were reported as ‘unique’.

Reverse transcription-polymerase chain reactions (RT-PCR), and sequencing reactions were performed using a commercial assay, ViroSeq^TM^ HIV-1 Integrase RUO Genotyping kit (VS-Int) (Celera Corporation, New Jersey, USA) as per manufacturer’s instructions and an ‘inhouse’ integrase assay (IH-Int) as previously described elsewhere [22]. Capillary electrophoresis was performed on an ABI 3130XL Applied Biosystems ^TM^ Genetic Analyzer. 

The raw sequence data from the sequencer were then assembled and edited using Sequencher^®^ version 5.0 DNA sequence analysis software (Gene Codes Corporation, Ann Arbor, MI, USA) and manually edited using BioEdit version 7.2.0 software [23]. Unique sequences covering the IN-codon positions 51–263 were included in the analysis. 

INSTI DRMs and accessory mutations were assessed using the Stanford University HIV drug resistance database algorithm version 8.7 (https://hivdb.stanford.edu/hivdb/by-sequences/ (accessed on 19 February 2021)). The mutations of interest included: T66A/I/K, E92Q, G118R, E138K/A/T, G140S/A/C, Y143R/C/H, S147G, Q148H/R/K, N155H and R263K [24]. Maximum-likelihood phylogenetic trees (bootstrap 1000) were generated using molecular evolutionary genetic analysis (MEGA) version X 10.1.6 [25] and included some duplicate sequences from the same individual collected over different time points (they cluster together with bootstrap values >95).

Determination of HIV-1 subtype was performed using Rega HIV subtyping tool v3.0 (http://dbpartners.stanford.edu:8080/RegaSubtyping/stanford-hiv/typingtool/ (accessed on 19 February 2021)) and the Stanford University HIV drug resistance database algorithm version 8.7 (https://hivdb.stanford.edu/hivdb/by-sequences/ (accessed on 19 February 2021)) [24,26]. 

We report reverse transcriptase (RT) and protease (PR) sequences for only the 11 patients found to have integrase DRMs. In instances where paired RT or PR sequences could not be performed, we looked for historical RT and PR results of the individuals and reported them. For one individual, we could not identify their paired or historic RT or PR sequences.

## 3. Results

We retrieved 78 residual plasma samples from May 2015–September 2019, representing 65 unique individuals failing either a DTG or RAL-based cART regimen at the time of genotyping request. Of the 78 plasma samples, 40 (51%) (from 34 individuals) were successfully amplified and sequenced for HIV-1 Integrase by VS-Int and IH-Int assays Figure 1. Basic demographics and VL results of these individuals are shown in Table 1. 

Amongst the 34 individuals, we obtained cART initiation dates of 24: their mean duration on ART was 11 years (Q1, Q3:8, 13) at the time of sampling for GRT.

All generated viral sequences were HIV-1C. Among the 34 individuals, 24 were on DTG ART, three on RAL cART and seven on ‘unknown’ INSTI (either DTG or RAL) at the time of collection of the sample for GRT. INSTI DRMs were found in 11(32%) of the study participants; their mean age (Q1, Q3) at GRT was 43 (40, 44) years, 7 (64%) of the 11 were male, and their mean duration on ART (Q1, Q3) was 11.5 (10, 13) years. Seven (64%) of the 11 individuals had documentation showing they were previously on a RAL-based regimen. The remaining four individuals were on a DTG regimen at sampling for resistance testing and had no known history of RAL cART exposure. The HIV-1 reverse transcriptase (RT) and protease (PR) region of the 11 individuals were sequenced and/or their historical RT and PR DRMs retrieved Table 2. 10 (91%), 8 (73%) and 4 (36%) of these 11 individuals had nucleoside/nucleotide reverse transcriptase inhibitor (NRTI), non-nucleoside reverse transcriptase inhibitors (NNRTI) and protease inhibitors (PI) DRMs respectively. 

One (9%), 6 (55%) and 4 (36%) had 2-class, 3-class and 4-class multidrug-resistant HIV. 7 (64%) of the individuals are currently virologically suppressed and 4 (36%) are not suppressed. Of the suppressed individuals, 6 (86%) have no PI DRMs and 1 (14%) has major PI DRMs. Six out of the 7 currently suppressed individuals are on a salvage regimen anchored by DTG and darunavir boosted by ritonavir (DRV\r) Table 2. Three of the four individuals who were not suppressed had extensive DRMs involving 4-classes of ARV drugs. One of the individuals not suppressed has recently demised. The distribution of INSTI DRMs amongst the 11 individuals is shown in Figure 2 and Appendix A. 

Amongst the individuals previously exposed to RAL ART (seven out of eleven), DRMs selected whilst failing their current DTG based regimens were E138K, G140A, Q148R; and N155H Figure 3. The four individuals failing DTG cART but with no documented prior exposure to RAL, their selected DRMs were E138K, G140A, Q148K, A128T; G118R, E138K; N155ND and T66A, G118R, E138EAKT Figure 3. 

## 4. Discussion

We evaluated virological, clinical, and HIVDR mutational pathways from a national programmatic cohort of highly treatment-experienced PLWH failing a DTG/RAL based regimen in Botswana. We identified 11 (32%) individuals with INSTI DRMs (eight being on DTG and three on RAL based regimens at HIVDR testing). They had a longer mean duration of prior cART at the time of GRT (11.5 years), and seven of them were previously exposed to RAL based regimens. Longer duration of ART has been previously associated with the development of HIV-DR [27]. Despite having INSTI DRMs, two out of the 11 individuals were on DTG once a day dosing; the Viking trials demonstrated twice a day dosing to be effective amongst such individuals [28]. 

A notable finding from our study was the identification of multi-drug resistance (MDR) HIV-1C variants; 2-class MDR 9.1%, 3-class MDR 54.5% and 4 class MDR 36.4%. We detected high rates of NRTI (90%), NNRTI (72.7%), and PI (36.4%) DRM’s. Another remarkable finding from our study was that despite this extensive MDR HIV-1C, nearly two-thirds of individuals (64%) are currently virologically suppressed. The Viking trial also showed 69% VL suppression despite extensive DRMs at 24 weeks [28]. The high VL suppression numbers despite multiclass DRMs could be explained by the inclusion of a potent PI-DRV\r and lack of PI DRMs in the suppressed individuals (six out of seven). It could also not be ruled out that DTG retains some suppressive activity against variants with some INSTI resistance mutations. It was recently shown that it is only upon the development of the T97A mutation that variants harboring Q148H and G140S Integrase mutations started to have increased VLs [22,27].

In our study, E138K (n = 5), S147G (n = 4), Q148R (n = 4) and N155H (n = 4) where the most frequent INSTI DRMs identified. This is in contrast to Y143C/R/H (n = 12), N155H (n = 9) and T97A(n = 13) from a similar study in the region where HIV-1C also predominates [29]. The longer duration of cART treatment of our participants with subsequent accumulation of mutations could explain these differences. Comparable with others, amongst individuals with prior exposure to RAL, the most common mutations selected were N155H (n = 4), Q148R (n = 4), S147G (n = 4) and E138K (n = 3) as others have similarly found [30,31].

Another unique finding of our study was the lack of selection of the integrase codon 263 substitution amongst the four individuals with virological failure who had no previous exposure to RAL cART. Compared with a similar cohort of patients from clinical trials, the selection of major integrase DRM- R263K was common [17]. Perhaps, this could be explained by the different HIV-1 subtypes; our cohort was all subtype C viruses whilst the other one was predominantly subtype B viruses.

Limitations of our study include the small sample size of participants; we managed to recruit 65 individuals and successfully genotyped the virus from 34 individuals. Although the study was heavily reliant on existing routine national programmatic clinical data, which often contains missing or unknown results, the results presented here represent ‘real-life’ events in HIV-1C diagnosed patients with prolonged cART treatment and DRMs. 

For the 68% of participants with VF while on DTG/INSTI based regimens but devoid of INSTI DRMs, we did not assess for other mutations in *nef*, *reverse transcriptase* and *protease* genes that could contribute to VF [32,33].

In summary, there is a need for frequent VL testing and/or genotypic resistance testing amongst treatment-experienced PLWH experiencing VF while on DTG-based regimens. In Botswana and most low/middle-income countries, DTG-based regimens are preferred as first and second-line anchor drugs amongst PLWH initiating and failing non-DTG based cART respectively [4,6]. Constructing effective ART regimens amongst treatment-experienced individuals with over 3-class DRMs in resource-limited settings can be daunting considering the limited drug therapeutic options, need for regular training to ensure correct dosing of ARVs and lack of genotypic and phenotypic resistance testing. Similarly, to what others have found [34,35,36], we recommend that boosted darunavir should be included as part of a salvage regimen amongst treatment-experienced individuals identified to have multiclass HIV DRMs. For individuals with multi-class DRM who are not virologically suppressed, optimizing ART regimens to include newer ARVs such as the attachment inhibitor-fostemsavir and/or anti-CD-4 antibody-ibalizumab might suffice and avert death [37,38]. In the era of mass roll-out of DTG based regimens, there is a need for continued surveillance for INSTI DRMs and determining the clinical significance of these mutations in HIV-1C infections.

## Figures and Tables

**Figure 1 viruses-13-00594-f001:**
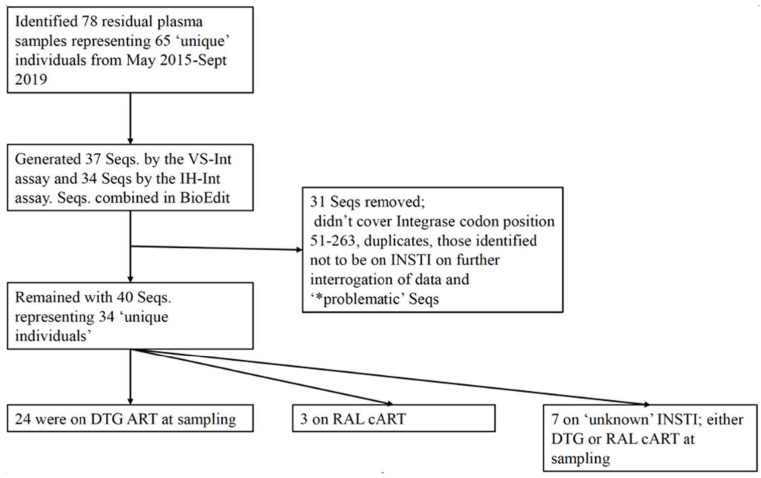
Flow diagram revealing the number of plasma, sequences and patients on various cART regimens. Seqs, sequences; DTG, dolutegravir; RAL, raltegravir, cART, combination antiretroviral therapy; INSTI, integrase strand transfer inhibitors. VS-Int, ViroSeq^TM^ HIV-1 Integrase RUO Genotyping kit Celera Corporation, USA; IH-Int, inhouse integrase assay. * Seqs clustering together with high bootstrap (>95) but being from different individuals.

**Figure 2 viruses-13-00594-f002:**
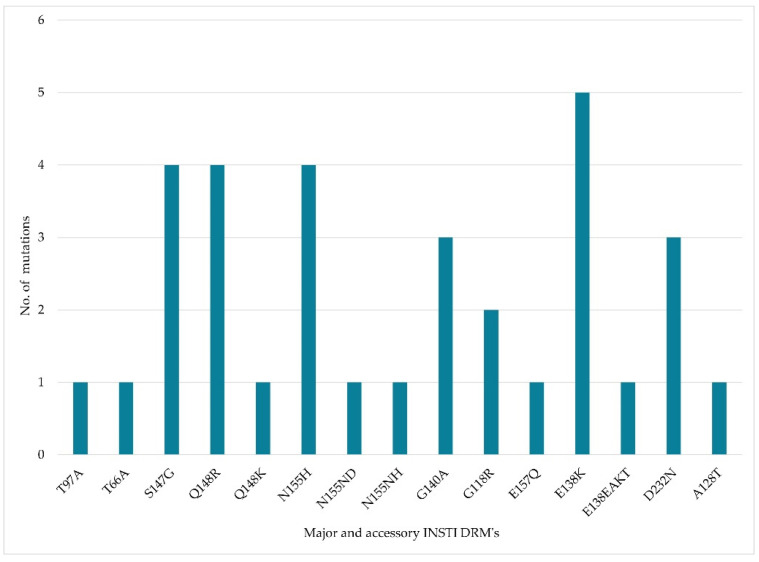
Prevalence of INSTI DRM in the 11 HIV-1C Integrase sequences from treatment-experienced individuals with virological failure whilst on dolutegravir/INSTI based therapy. INSTI, integrase strand transfer inhibitor, DRMs drug resistance mutations.

**Figure 3 viruses-13-00594-f003:**
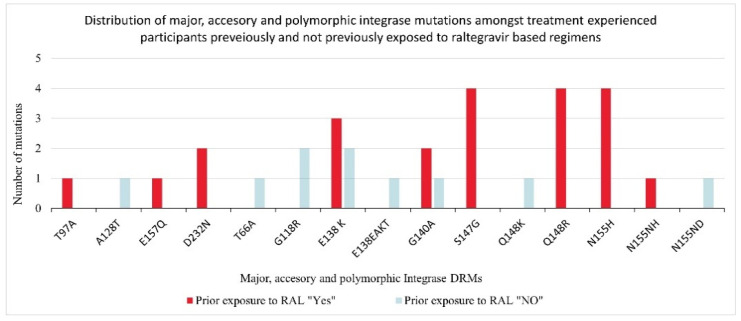
Distribution of integrase drug resistance mutations according to prior exposure to raltegravir. DRM, drug resistance mutations, RAL, raltegravir.

**Table 1 viruses-13-00594-t001:** Basic demographics and viral load characteristics of individuals failing DTG/RAL cART.

Basic Characteristics		All Participants (n = 65)	Participants with Sequences Generated (n = 34)
* Age (years), median (Q1, Q3)		40 (27, 49)	41 (26, 45)
Gender	Female n (%)	37 (57%)	15 (44%)
	Male n (%)	28 (43%)	19 (56%)
Median log _10_ HIV-1 VL (Q1, Q3) copies/mL		^†^ 3.78 (2.13, 4.49)	^±^ 4.53 (3.98, 5.10)

N.B; VL was measured from all plasma samples sent for GRT before commencing testing as per standard of care. ^†^ VL written as <20, <400 or <40 or TND, we used 20, 400, 40 and 20 respectively for analysis. In this analysis they were nine out of 65. * 6 individuals did not have age written on the requisition and could not calculate their age.; they are not included in the analysis of age. ^†±^ VL was from 35 of the 40 plasma samples at the time of GRT as 5 had UNK VLs. cART, combination antiretroviral therapy; DTG, dolutegravir; RAL, raltegravir; VL, viral load, UNK, unknown; TND, target not detected.

**Table 2 viruses-13-00594-t002:** Demographics, viral loads, cART regimens and DRM’s of 11 individuals with multi-class drug resistance in Botswana.

Patient #	ART Initiation Date	Date Sample Collected	Age at Sample Collection	Gender	VL of Sample Collected for GRT (cps/mL)	ART Regimen at Time of Sampling of GRT	Prior Exposure to RAL (Yes/No)	Major DRM’s ^Β^			Current ART Regimen	Current VL (cps/mL)	Date of Current VL
								RT(NRTI; NNRTI)	PI	INSTI			
1 (139-0007-8)	27-Apri-06	31-May-17	40	M	69,510	TDF/FTC/DRV\r/DTG	NO	D67N, K70R, M184V, T215I, K219E; A98G, V106I, Y188L	NONE	E138K, G140A, Q148K, (A128T)	TDF/FTC/DRV\r/DTG 200/300 mg od/600\100 mg bd/50 mg bd	<400	10-Sep-20
2 (139-0006-2)	25-Nov-03	21-Jun-17	38	F	826	TDF/FTC/DTG	YES	M184V; K101E, G190S	NONE	Q148R	TDF/FTC/DTG 200/300 mg OD/50 mg BD	<50	22-Jul-20
3 (139-0008-6)	21-Sept-11	^±^ 10-Jan-18	43	M	* <400	AZT/3TC/DTG	NO	A62V, K65R, M184V; K103N, V106M	NONE	G118R, E138K	3TC/DRV\r/DTG (150 BD/600\100 BD/50 OD)	<25	02-Sep-20
^∞^ 4a (139-0009-5a)	14-May-07	19-Apr-17	41	F	119,563	ABC/3TC/DTG	YES	M184V, T215Y; K103S, G190A	NONE	S147G, N155H, D232N	TDF/FTC/DRV\r/DTG (300/200 mg od/600\100 mg bd/50 mg bd)	<25	11-Jun-20
^∞^ 4b (139-0009-5b)		^±^ 18-May-18	42	F	79,028	ABC/3TC/DTG	YES	M184V, T215Y; K103S, G190A	NONE	S147G, N155H, D232N	TDF/FTC/DRV\r/DTG (300/200 mg od/600\100 mg bd/50 mg bd)	<25	11-Jun-20
5 (139-0147-0)	27-Nov-06	28-Jun-17	21	M	815	TDF/FTC/DTG	NO	ND	ND	N155ND	TDF/FTC/DTG 300/200/mg OD/50 mg BD	12,304	24-Jul-19
6 (139-0002-8)	22-Sept-03	01-Nov-17	51	M	515	TDF/FTC/DRV\r/DTG	YES	*+ K70R, M184V; K219N/Y181C 20APRIL2009) * ¥ D67N, K70R, M184V/ NONE (18AUG2016)	*+ V32I, I47V, I54L, I84V (20 APRIL2009) * ¥ V32I, I47V, I54L, I84V (18AUG2016)	E138K, G140A, S147G, Q148R, (T97A)	TDF/3TC/DRV\r/DTG (300/300 mg OD/600\100 BD/50 mg BD)	22,690	31-Aug-20
7 (139-0004-6)	19-May-04	06-Apr-18	45	M	* 55,342	AZT/3TC/DTG	NO	M41L, T69G, K70R, M184V, T215C, K219E; A98G, K101E	M46I, I54V, L76V, V82A	T66A, G118R, E138EAKT	TDF/FTC/DRV\r/DTG 300/200 mg OD/600\100 mg BD/50 mg OD	992	13-Oct-20
8 (139-0003-4)	25-June-04	11-Apr-18	41	F	50,699	TDF/FTC/RAL	YES	M184V, T215Y; NONE	NONE	E138K, G140A, Q148R	TDF/3TC/DRV\r/DTG 300/300 mg OD/600\100 mg BD/50 mg BD	<25	22-Jul-20
9 (139-0001-8)	7-May-01	06-Dec-18	55	M	9775	TDF/3TC/DTG	YES	M41L, D67N, K70KR, V75M, M184V, L210W, T215Y, K219E; A98G, Y181C, G190A	M46I, I47V, I54L, L76V, I84V, Q58E, N83D	E138K, S147G, Q148R, N155H, (E157Q)	TDF/3TC/DTG	177,268	13-Feb-20
10 (139-0011-3)	6-Feb-04	13-May-15	50	F	1300	TDF/FTC/DRV\r/RAL	YES	**+ D67N, K70R, K219E/K101Q, K103N (7DEC2009) ** ¥ NONE/P225H (10FEB2012)	**+ NONE(7DEC2009) ** ¥ NONE (10FEB2012)	N155H	TDF/FTC/DRV\r/DTG 300/200 mg OD/600\100 mg BD/50 mg BD	<400	10 Jan 2020
11 (139-0012-9)	3-Oct-02	18-May-15	44	M	* 2300	TDF/FTC/DRV\r/RAL	YES	Major; M184V, M41L, T215Y: NONE	Major; M46I, V82A Accessory; L24I	N155NH (D232DN)	AZT/3TC/DRV\r/DTG 450 mg BD/600\100 mg BD/50 mg BD	<50	26-Aug-20

^Β^ Major DRMs as determined by the Stanford HIV drug resistance database. ± Date sample collected not written on requisition forms, used date sample received in a testing laboratory and/or date samples testing started and/or results issued. * No VL results available for the sample, used last recorded VL before sampling for GRT obtained from IPMS. Historical DRMs denoted with a ’*’. Historical DRMs refers to GRT performed on plasma samples from the same individuals but at a different time point (date in appendix table). *^+^ from sample GRT performed on 20 April 2009. * ^¥^ from sample GRT performed on 18 Aug 2016. **^+^ from sample GRT performed on 7 DEC 2009. ** ^¥^ from sample GRT performed on 10 FEB 2012. ∞ 4a and 4b are the same individual but their specimens were collected at different time points. Major INSTI DRMS column, DRMs listed within brackets “()” are accessory INSTI resistance mutations. VL, viral load; ART, antiretroviral therapy; GRT, genotypic resistance testing; cps/ml, copies/mL; RT, reverse transcriptase; NRTI, nucleoside/nucleotide reverse transcriptase inhibitors; NNRTI, non-nucleoside reverse transcriptase inhibitors; PR, protease; PI, protease inhibitor; INSTI, integrase strand transfer inhibitors; DRMs, drug resistance mutations; VL, viral load; GRT, genotypic resistance test; RAL, raltegravir; 3TC, Lamivudine; DRV\r, darunavir\ritonavir; DTG, dolutegravir; FTC, emtricitabine; TDF, tenofovir disoproxil fumarate; BD, twice a day dosing; OD, once a day dosing; mg, milligrams; ND, not done; IPMS, integrated patient management software (a laboratory information systems software). Green colour depicts virological suppression and red colour non virological suppression.

## Data Availability

IN sequences have been deposited into the national centre for biotechnology information (NCBI) GenBank and their accession numbers are MW690052-MW690089, MG989443.1, MG989444.1.

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
