# Peer review of "HIV-1 Subtype C Drug Resistance Mutations in Heavily Treated Patients Failing Integrase Strand Transfer Inhibitor-Based Regimens in Botswana†"

_viruses, 2021, doi:10.3390/v13040594_

Round 1

Reviewer 1 Report

In this study, the authors have investigated the mutations in HIV-1 C which may be responsible for the lack of therapeutic efficacy in dolutegravir-based regimens. This is a novel and interesting study and will undoubtedly add to our current apprehension about the molecular mechanisms of resistance to DTG in patients. There are a few issues that need to be addressed by the authors in order to improve their study:

1-There are a few typos and grammatical mistakes throughout the manuscript that need to be corrected.

2-Was the informed consent obtained from the patients? Was the study was approved by any Ethical Committee in Botswana? This critical information needs to be clearly written in the relevant section of the manuscript.

3- Reverse transcriptase-polymerase chain reactions (RT-PCR) should be replaced by quantitative reverse transcription-PCR (qRT-PCR).

4-Lines 134-136, please insert the proper citation and remove “Error! Reference source not found”. The same with line 160 and other lines throughout the manuscript.

5-A more concise title could be “Drug resistance promoting mutations in HIV-1 C patients in Botswana”.

Reviewer 2 Report

In this work authors found 78 samples from HIV patients on cART and through gene sequencing they found drug resistant mutations which are cause of lack of suppressing HIV. This analysis is very useful for patients with DTG and RAL who are not able to reduce their viral loads, so that they can be moved other treatments if certain mutations were identified.

What is the main question addressed by the research?

What mutations are reason for Drug resistance

Is the paper well written?

Yes, certain areas can be improved by rewritting.

Is the text clear and easy to read?

Yes, certain areas can be improved by rewritting.

Are the conclusions consistent with the evidence and arguments presented?

yes

Do they address the main question posed?

yes

Round 2

Reviewer 1 Report

The authors have successfully addressed my comments and I have no further comment/question to raise.